# Diesel in Antarctica and a Bibliometric Study on Its Indigenous Microorganisms as Remediation Agent

**DOI:** 10.3390/ijerph18041512

**Published:** 2021-02-05

**Authors:** Rasidnie Razin Wong, Zheng Syuen Lim, Noor Azmi Shaharuddin, Azham Zulkharnain, Claudio Gomez-Fuentes, Siti Aqlima Ahmad

**Affiliations:** 1Department of Biochemistry, Faculty of Biotechnology and Biomolecular Sciences, Universiti Putra Malaysia, UPM Serdang, Selangor 43400, Malaysia; msrasidnie21@gmail.com (R.R.W.); syuenylim@gmail.com (Z.S.L.); noorazmi@upm.edu.my (N.A.S.); 2Department of Bioscience and Engineering, Shibaura Institute of Technology, College of Systems Engineering and Science, 307 Fukasaku, Minuma-ku, Saitama 337-8570, Japan; azham@shibaura-it.ac.jp; 3Department of Chemical Engineering, Universidad de Magallanes, Avda. Bulnes, Punta Arenas, Región de Magallanes y Antártica Chilena 01855, Chile; claudio.gomez@umag.cl; 4Center for Research and Antarctic Environmental Monitoring (CIMAA), Universidad de Magallanes, Avda. Bulnes, Punta Arenas, Región de Magallanes y Antártica Chilena 01855, Chile; 5National Antarctic Research Centre, Universiti Malaya B303 Level 3, Block B, IPS Building, Kuala Lumpur 50603, Malaysia

**Keywords:** Antarctica, diesel, bioremediation, microbial degradation, psychrophiles

## Abstract

Diesel acts as a main energy source to complement human activities in Antarctica. However, the increased expedition in Antarctica has threatened the environment as well as its living organisms. While more efforts on the use of renewable energy are being done, most activities in Antarctica still depend heavily on the use of diesel. Diesel contaminants in their natural state are known to be persistent, complex and toxic. The low temperature in Antarctica worsens these issues, making pollutants more significantly toxic to their environment and indigenous organisms. A bibliometric analysis had demonstrated a gradual increase in the number of studies on the microbial hydrocarbon remediation in Antarctica over the year. It was also found that these studies were dominated by those that used bacteria as remediating agents, whereas very little focus was given on fungi and microalgae. This review presents a summary of the collective and past understanding to the current findings of Antarctic microbial enzymatic degradation of hydrocarbons as well as its genotypic adaptation to the extreme low temperature.

## 1. Introduction

The ever-increasing Antarctic expeditions have rendered the environment more vulnerable to diesel pollution. Due to its remoteness, one of the notable challenges in sustaining human presence in Antarctica is the enormous amount of energy required to empower transportation, research stations and activities concentrated at the coastal ice-free rocky areas. This energy demand is typically met by producing energy with diesel combustion. The transportation and long-term storing of fuels in bulk quantity have elevated the occurrence of contamination [1]. The most common diesel contamination in Antarctica is descended from leakages during refuelling from a ship to a land-based facility, unmaintained fuel storage or pipe infrastructure and minor accidents in fuel handling processes. In addition to that, the extreme climate in the Southern Ocean has caused boating and aviation accidents, which also led to fuel leakages in the Antarctic environments [2].

Given that human activities in Antarctica are prospected to intensify and diversify, the Antarctic environmental conservation is pressured. The utmost challenge in protecting the polar environment is the effective management of hydrocarbon contaminants emerging from oil spills. Diesel pollutants tend to be more persistent in the subzero temperature compared to those of warmer climes. This is due to decreased volatility and evaporation at low temperature, amalgamated with innate nutrient limitations that significantly inhibit metabolic activity of hydrocarbon-degrading microbes [1,3]. Policies, attitude and awareness on the environmental protection have improved in recent years; however, the risk of diesel pollution remains. If left untreated, the toxicity of diesel will cause negative effect on the living organisms in this polar region. Several studies have reported the toxicity effect of diesel fuel on the species richness, evenness and phylogenetic diversity on Antarctic and sub-Antarctic soil biota [4]; retarded germination and early growth of sub-Antarctic plants [3]; and lethal or behavioural effects on Antarctic aquatic organisms [5,6].

The Antarctic environment possesses significant intrinsic values that hold a great potential in bridging a large scientific and historic knowledge gap [7]. This remote region has been deemed worthy of protection and is presently under the scrutiny of several nongovernment organisations. After the Madrid Protocol was established in 1991 under the Antarctic Treaty System (ATS), Antarctica has been designated as a “natural reserve, devoted to peace and science” [8]. Among the content of this agreement is the Protocol on Environmental Protection to the Antarctic Treaty. Article 8 of the protocol requires all activities to go through prior environmental assessment, whereas Article 5 necessitates the inauguration of environmental monitoring procedures [7,8,9]. The current Antarctic regimes encompass the Antarctic Treaty Consultative Parties (ATCPs) and Antarctic Treaty Consultative Meeting (ATCM) as decision makers; whereas Committee for Environmental Protection (CEP), Conservation of Antarctic Marine Living Resources (CCAMLR) and Antarctic and Southern Ocean Coalition (ASOC) as special subsidiary bodies that govern the implementation of the Antarctic Treaty [7,10]. Other international organisation such as United Nations and Scientific Committee on Antarctic Research (SCAR) also participate in the environmental protection policy by further processing the transparency, legitimacy and effectiveness of the Antarctic governance [10]. 

Under the Madrid Protocol, all treaty nations are obligatory to establish contingency plans for accidental fuel spills as well as to minimise pollution unless the removing of contaminants can cause more detrimental effects to the environment [8]. Nonetheless, fuel management and contaminants regulation are solely dependent on self-regulation by national operators and their interpretation of the protocol [11]. Moreover, the availability of machinery and facilities are limited, and constructions of such are also restrained by virtue of the frigid temperature; therefore, the catabolic capacities of indigenous microorganisms capable of degrading hydrocarbons has emerged as one of the most important tools to eliminate or reduce diesel contamination in Antarctica [6,12,13]. This paper was aimed at reviewing diesel uses in Antarctica, hydrocarbon pollution, its toxicity as well as the roles and adaptation of microbes in the polar climate for remediating diesel along with its bibliometric analysis.

## 2. Review Methodology

This review presents the dependency of diesel in Antarctica and its pollution effect on the environment, as well as bioremediation by Antarctic indigenous microorganisms, that was supported by a bibliometric study as a systematic approach. Research documents from the time phase 1990 until 2020 were retrieved from the Scopus database using the following keywords: diesel OR hydrocarbon * AND biodegradation OR bioremediation OR phytoremediation OR “microbial degradation” AND Antarctic * filtered in article title, abstract and keywords, which resulted in 207 documents. Biblioshiny was used to analyse the annual and country scientific production; while VOSviewer was chosen as a visualisation tool for the co-occurrence keyword analysis.

## 3. Diesel and Its Use in Antarctica

Diesel oil is a petroleum product that supplies fuel for automobiles, aircraft and ships, as well as generates electricity in some areas to power homes and commercial buildings. According to Speight [14], diesel fuel is one of the middle distillates of petroleum with the boiling point of 175 °C to 375 °C in approximate, which is higher than gasoline but not more than gas oil. The full composition of diesel fuel is complex, varied and beyond the scope of this report. Nevertheless, the main components of diesel include 25% to 50% of alkanes, 20% to 40% cycloalkanes and 10% to 57% of aromatic contents [15,16]. Diesel fuel also contains a trace amount of polar organic compounds including sulphur, nitrogen and oxygen, as well as metals such as nickel, vanadium, iron and copper [17]. 

As mentioned, diesel is extensively used to complement human activities in Antarctica. Traditionally, heavy and intermediate grade fuel oils are used for larger vessels to operate in Antarctic waters; however, in response to concerns on the persistence of contaminants in the case of oil spill, International Maritime Organisation (IMO) implemented a ban on the use of heavy fuel oil (HFO) in water bodies governed by the Antarctic Treaty [17]. Subsequently, the heaviest grade fuel oil that is currently allowed for large vessels is the intermediate grade fuel oil (IFO 180), except for government vessels or those that are engaging in search and rescue operations [2,18]. On the other hand, most research stations in Antarctica use Special Antarctic Blend (SAB) diesel, which is exclusively blended for better performance in cold temperature. SAB has a lower heating value of 35,274 kJ/L, a density of 0.805 kg/L at 15 °C and sulphur content of 10 mg/kg max [19]. 

As encouraged by The Antarctic Treaty System, many Antarctic research stations are currently opting for the use of renewable energy sources like solar, ocean and wind energy. However, most stations in the region still require the combustion of diesel and other hydrocarbons to accommodate their power requirements. This is mainly due to the complications regarding wind and ocean turbines or the absence of sunlight during certain seasons in the region. Furthermore, most nations are still in early stages of research in enacting the use of renewable energy in their Antarctic research stations [20,21,22]. To date, only Belgium’s Princess Elisabeth Antarctica Research Station has been reported to be fully depending on renewable energies to meet its energy requirement [23]. A collective summary of reported approximate annual fuel consumption and endeavours on renewable energies in some Antarctic research stations is shown in Table 1. Most renewable energy undertakings in Antarctica focus on reducing instead of eliminating fuel consumption. While this might conserve the polar environment, it is unlikely to significantly reduce diesel pollution risk.

## 4. Diesel Pollution and Toxicity

Unlike in the North Pole, there is no permanent human population in Antarctica since the inhabitants are mostly tourists, scientists or support staff of the Antarctic research programs. Hydrocarbon contaminations are very common throughout the continent, especially areas close to research stations where anthropogenic activities are mainly conducted. Consequently, high concentration of petrogenic hydrocarbon contaminants and their biomarkers have been reported in soil and surface sediment samples close to fuel tanks and boat storeroom where fuel combustion and storing have occurred [34,35]. These contaminants can seep into porous soils up to the barrier of impermeable permafrost, while slowly migrating with every summer melting, releasing the petroleum products to the surrounding area and eventually reaching the marine environment [36]. One of the most widespread environmental impacts of human activities in Antarctica occurred in King George Island, South Shetland Islands which contains nice operating research stations. This island is severely polluted due to poor fuel management [37,38]. In 1990, a huge amount of oil has leaked from storage tanks at Marsh Station, which apparently was an unintended issue for an ongoing period [37]. Another major oil spill from storage tank has been recorded in 2009 where an unknown amount of diesel fuel was discharged into the snow, which then rippled into Maxwell Bay during snowmelt [38]. In winter 2015, a pipeline has spilled 4000 L of SAB diesel fuel in Casey Station, Antarctica; however, due to heavy snow, the contaminated site was unseen, therefore covering the spreading of the fuel spill for more than three months [39]. 

Another main source of diesel pollution, particularly in the marine and coastal areas comes from boating accidents. A detailed review of oil spillage incidents into the marine environment was reported by Lim et al. [40]. Hitherto, the most significant oil spill incident in Antarctica water is the sinking of Bahía Paraíso in Arthur Harbour, Antarctica in January 1989. This tragedy had released 600,000 L of diesel fuel arctic (DFA) into the Southern Ocean due to a 10-m tear in the ship’s hull [41]. Aside from that, in the last two decades, the number of cruise ships voyaging the Atlantic Ocean has escalated [42]. Inevitably, an increase in accidents involving cruise ships has also been reported. This has been a growing concern to the Antarctic and Southern Ocean Coalition (ASOC) as well as members of the Antarctic Treaty as the conservation of the Antarctic region had been jeopardised [43]. In February 2007, the hull and fuel tanks of M.S. Nordkapp have been damaged, causing the vessel to ground at Deception Island spilling around 1000 L of diesel [44,45,46]. Later that year, cruise ship M.S. Explorer was holed by ice causing 189,271 L of diesel consequently leaked into the ocean [47,48]. 

The pristine and sensitive environment of the Antarctic continent prompts a significant impact of hydrocarbon pollution, even at a local level [34]. Hydrocarbon contaminants have been proven to be long-lived in the region, except in high-energy marine environments where dispersion occurs rapidly [49]. Furthermore, the low temperatures and abundant snow in some regions favour the atmospheric deposition and persistence of hydrocarbon pollutants in soils and vegetation besides delaying the biological degradation of hydrocarbons in contaminated soils due to slower mitochondrial function and higher lipid storage of organisms, [34,50,51,52]. In addition, response to oil spill in Antarctica is constrained due to extreme weather and remote location, causing the clean-up procedures to often be delayed. For instance, after the grounding of Bahía Paraíso, the emergency spill-response team organised by the United States National Science Foundation (NSF) took 10 days to arrive with fuel-spill response equipment for initial survey and clean-up [41].

Diesel toxicity is a great concern in Antarctica due to the aforementioned polar environmental factors. Petrogenic contaminants’ toxicity effects toward different organisms in the region have been widely reported [2,17,44,45]. Bacterial community commonly serves as an indicator of diesel fuel toxicity in the sub-Antarctic soil [4]. Soils in polar climates usually have tiny amounts of natural attenuation capacity, allowing contaminants to persist longer than they could in warmer environments [39]. Upon the occurrence of diesel fuel contamination, the total species richness, diversity and evenness in the Antarctic soils significantly declined, while the population of hydrocarbon-degrading bacteria, especially *Pseudomonas*, *Rhodococcus* and *Parvibaculum* drastically increased [4,53]. Oil-contaminated soils are also reckoned to be able to trap more heat due to the decreased surface albedo and that their hydrophobicity might increase, subsequently decreasing soil moisture holding capacity [54]. On the contrary, spilled oil also shrank the species diversity of microorganisms in the marine environments, where population is dominated by *Oleisprisa*, *Granulosicoccus*, *Cycloclasticus* and *Rhodococcus* genus [55,56,57]. 

Penguins and Antarctic seabirds are also heavily affected by diesel pollution. Most reports of diesel toxic effects on birds are of the aftermath of the grounding of Bahia Paraiso. Oiled Adelie penguins and blue-eyed shags have been found dead in Biscoe Bay which is adjacent to the grounding site [58]. Additionally, Eppley [59] reported a total reproductive failure of South Polar Skuas that occurred at Palmer Station and the death of all skua chicks in the local population. Diesel also presents indirect effect on seabirds specifically through food webs, where poisoning might occur due to hydrocarbon-contaminated fishes or changes in their prey and predator’s population [59]. Most vessels that sunk in the Southern Ocean occurred in close proximity within foraging ranges of breeding penguins from their colony such as those at Cape Hallett, Ross Sea and Nightingale Island, and oil leakage from a pipe in Marsh Station that had caused contamination in Elephant Valley, an important breeding area from some birds and penguin species [37,44]. Penguins are known to have a great capacity of lipid storage. Brown [17] explained that while this higher lipid content can provide initial protective effect to oil contaminants, diesel is mostly hydrophobic, therefore can be stored in the lipid of these species contributing to more toxic effects. 

Apart from that, invertebrates of Antarctica are heavily affected by diesel contaminations. The soil invertebrate community is mostly dominated by nematodes worms, rotifers and tardigrades, whereas marine invertebrates include bivalves, echinoderms and isopods [2,17,60]. Due to their physiochemical and ecological adaptations to extreme environment, soil and marine invertebrates might be the most affected compared to other classes of livings [17]. In 1987, the grounding of Nella Dan had affected many marine invertebrates where they were found washed up dead along the shoreline of Macquarie Island [61]. Diesel has also been proven to be mutagenic to several species of isopods and springtails and shown to limit reproduction rate in annelid worms [2]. In addition, among the lifecycles of marine invertebrates such as bivalves and echinoderms, it has been found that embryonic and larval stages are more affected by oil due to the rapid cell growth and differentiation [17]. Heretofore, the sensitivity estimates of diesel pollutants that may affect these Antarctic communities are still not adequately assessed.

## 5. Role of Antarctic Microbes in Diesel Remediation—A Bibliometric Analysis

Presently, bioremediation has emerged as a superior and the most ecofriendly approach in removing or neutralising contaminants [62,63]. As mentioned, the remoteness of Antarctica has led to limited availability of machinery and facilities. Thus, response to accidental oil spill tends to delay or is absent. In this context, the catabolic capacities of Antarctic indigenous microorganisms hold a great potential in removing contaminants while leaving minimal or no side effects [12]. Antarctic hydrocarbon-degrading microorganisms have optimum enzymatic activity in low temperature and could excellently adapt to the polar temperature [64]. Natural attenuation potential of autochthonous microorganisms has been proven but very limited [65]. Therefore, several types of bioremediation techniques have been applied at the region to accelerate biodegradation rate, thus minimising the toxicity effect of diesel fuel to the indigenous organisms. These methods, along with their feasibility and limitations, are shown in Table 2.

A bibliometric analysis on microbial diesel remediation in Antarctica was conducted to analyse the growth or lack thereof of research attention of this topic. In the last three decades, an annual growth of 9.97% has been observed in the publication with the keyword query mentioned in Section 2. The increasing trend indicates the growing attention of microbial hydrocarbon remediation in the South Pole region (Figure 1). Data earlier than 1990 was not found from the search, and the annual production was put into groups of a decade to avoid flatness. The steep rise in the number of publications in 2001 until 2010 coincided with the enforcing of the Environmental Protocol of the Antarctic Treaty in 1998 that was meant to enhance the protection of the Antarctic environment [75].

Upon further breakdown of literature growth, it has been found that the top 10 most productive countries in this research theme are Australia, Argentina, United States of America (USA), Malaysia, Italy, Canada, United Kingdom (UK), Brazil, France and New Zealand. The pie chart depicts each country’s domination in the research of microbial hydrocarbon remediation (Figure 2). Other countries with more than 10 publications that are not shown in the figure are China, Germany, Chile and Japan. This also coincides with the Antarctic Treaty as all the countries mentioned are the members of the Environmental Protocol. This indicates the continuous effort of researchers in preserving the Antarctic environment, which indirectly implies the important role of Antarctic indigenous microbes in the cleaning-up of fuel pollutants in the region as the protocol requires contaminated sites to be cleaned after all expeditions. This analysis also suggested that bioremediation is the preferred method in keeping the environment clean while adhering to the protocol.

The studies on hydrocarbon bioremediation in Antarctica have mostly focused on bacteria. This was concluded upon analysing the keywords in these publications, which implied their research themes. The co-occurrence networks of the most relevant keywords are shown in Figure 3. The visualisation map depicts the thematic focus in the field of research by grouping keywords into different clusters, which are colour-coded. A total of eight clusters can be observed showing a diverse range of research focus using varies microorganisms in bioremediation. The size of the circle represents the frequency of the keyword co-occurrence, whereas the thickness of the line indicates the strength of relevance. As illustrated, the keyword “bacteria (microorganisms)” was the largest as compared to other microbes, which indicates the highest frequency of co-occurrence. The commonly used hydrocarbon-degrading bacteria include *Rhodococcus*, *Pseudomonas* and *Arthrobacter* [53,76]. Other important keywords shown in the network visualisation aside from bacteria and its derivative keywords are “fungi” and “algae”, which demonstrated the feasibility of performing bioremediation using microbes besides bacteria.

## 6. Microbial Diesel Degradation and Its Metabolic Pathways

As depicted in Figure 3, articles on microbial remediation in Antarctica were centred on bacteria. Bacteria are the most favoured microbes in hydrocarbon degradation due to their versatility and ability to utilise diesel as their sole carbon source while leaving no side effects [77]. The feasibility of microbial remediation relies heavily on the optimised process parameters. Therefore, plenty of studies involving optimisation of diesel degradation have been documented. These reports on hydrocarbon-degrading bacteria in low temperature as well as their brief research summaries are shown in Table 3. Based on this finding, it can be concluded that most Antarctic bacterial species have the highest enzymatic activity at temperature lower than 25 °C, noting them as either psychrophilic or psychrotolerant.

Bioremediation using fungi and microalgae is not heavily documented as compared to that using bacteria. Several Antarctic fungal species have been characterised to have tolerance or ability to degrade hydrocarbons as shown in Table 4. Hitherto, the identification on Antarctic microalgal species hydrocarbon remediation is yet to be documented. Nonetheless, reports discussing the potential of microalgae in degrading petroleum hydrocarbons have been found. The hydrocarbon-degrading ability was postulated through the presence of hydrocarbon degradation intermediates such as propanoic acid, decanoic acid and dodecanoic acid [83,84]. Moreover, Antarctic terrestrial alga *Prasiola crispa* was reported to be tolerant of SAB fuel-contaminated soil, implying the ability of the alga in assimilating the hydrocarbon contaminants [85]. 

The metabolic pathways of bacterial and fungal hydrocarbon degradation are varied; nonetheless, one critical division of mechanisms is based around aerobic degradation [92]. Biodegradability of oil components increase in the order of polycyclic aromatic hydrocarbons (PAHs), cyclic alkanes, monoaromatics, low molecular weight n-alkyl aromatics, branched alkenes, branched-chain alkanes and n-alkanes [93]. The degradation of paraffin, which comprises straight or branched alkanes, can be classified as terminal, diterminal or subterminal [94]. Generally, these microbes catabolise diesel into simple organic compounds such as alcohols, acids and fatty acids, which will eventually undergo β-oxidation to form acetyl-CoA [1]. The general pathways of alkane degradation are shown in Figure 4.

Cyclic alkanes are relatively more resistant to microbial attack since they lack an exposed terminal methyl group [94]. Several studies have shown a few species of temperate bacteria that are able to use cyclic alkanes as their sole carbon source by utilising the alkyl side chain, thus initiating breakdown process [97,98]. Cyclic alkanes are first converted to cyclic alcohols, which then undergo dehydrogenation to ketones before finally opened by the action of lactone hydrolase [95]. The documentation of naphtene degradation mechanism with fungi was not found. On the other hand, aromatic hydrocarbons are another major compound of diesel. Due to its nature state, the degradation of aromatic hydrocarbons is more complex compared to alkane and cycloalkane [96]. Generally, there are two major steps in the degradation of aromatic molecules namely the activation of the ring and ring cleavage. The activation of ring is achieved by the presence of two molecular oxygen atom incorporating into the aromatic ring and leads to the dihydroxylation of aromatic nucleus, producing a less stable compound [99]. This compound is then further oxidised to catechol, which serves as a precursor to ring cleavage. Then, a series of reactions involving several enzymes will eventually produce acetyl-CoA, leading to the formation of energy for the microbes involved [100]. The bacterial and fungal mechanisms of aromatic hydrocarbon degradation are shown in Figure 5.

## 7. Microbial Adaptation Mechanisms in Cold Region

Microbes are capable in adapting to a wide variety of environments because of their diverse metabolisms [101]. In a review by Sarmiento et al. [102], the authors demonstrated the adaptation of cold-active enzymes in maintaining high flexibility and activity in frigidness. The adaptation includes changes in core and surface hydrophobicity, alterations of amino acid compositions, weaker protein interactions, decreased secondary structures and oligomerisation and increased conformational entropy of unfolded proteins. Microorganisms have astonishing adaptation to withstand extreme conditions of their surroundings [103]. Particularly in Antarctica, indigenous microbes are obligated to cope with subzero temperatures, temperature fluctuations, frequent freeze and thaw process as well as intense solar irradiations [104,105]. 

As a response to the cold-stress, Antarctic bacterium *Flavobacterium* spp. was found to produce more major fatty acids in its cell membrane compared to the mesophiles and psychrophiles of this genus [106]. In addition, Antarctic psychrophiles *Planococcus* sp. NJ41 and *Shewanella* sp. NJ49 have been found to contain abundant β-carotene in the cells, which is known to be the key mediator of membrane fluidity in cells, thus explaining the ability of Antarctic bacteria in excellently adapting to the extreme low-temperature [79,107]. In the context of hydrocarbon remediation, a reported characteristic of cold-adapted hydrocarbon-degrading bacteria is the prevalence of alk genes in *Rhodococcus*, *Pseudomonas* and *Acinobacter* species [108,109]. These genes, particularly the *alkB* genes, are known to encode for alkane monooxygenase, which is a key enzyme in the degradation of alkane. The *alkB* genes are commonly used as a functional biomarker for the characterisation of aerobic alkane degradation [110]. Furthermore, the *nahAC* genes that are closely related to *Pseudomonas*, which code for the breakdown of naphthalene, have been also detected in hydrocarbon-polluted soil community from Carlini station and Potter Peninsula [111].

Commensurable to bacteria, fungi are ubiquitous in nature and possess specific mechanisms to thrive in harsh environments [112]. A comprehensive list of adaptation strategies of psychrophilic fungi to cold-stress has been reported in Buzzini et al. [113]. The contribution of cold-adapted yeasts in hydrocarbon remediation is crucial under certain conditions especially when bacterial growth is inhibited as they are usually more resistant to UV radiation, alterations of pressure and salinity [114,115]. Antarctic isolated yeasts *Pichia caribbica* and *Guehomyces pullulans* have been found to have cold-active amylase, cellulases, lipase, esterase, protease, pectinase and xylanase [116]. Although the correlation of these enzymes to hydrocarbon-degrading ability was not specified, it proved that these cold-adapted yeasts thrive in cold environment, therefore making them an important prospect in the growing research of polar biotechnology. 

Limited amount of microalgal genetic adaptations specific to the cold environmental circumstances have been documented. Routaboul et al. [117] mentioned that psychrophilic autotrophs maintain a high content of trienoic fatty acid, which is important in ensuring proper biogenesis and chloroplasts maintenance in long-term growth at low temperature. The development of high fluidity biological membranes to avoid loss of ion permeability is also crucial to ensure optimal photosynthetic function in the polar environment [118]. More recently, an unusually large number of genes encoding for light harvesting complex (LHC) protein has been also found in the gene ontology analysis of Antarctic diatom *Fragilariopsis cylindrus* when compared to temperate diatom [119].

## 8. Conclusions

The toxicity of diesel in Antarctica has been an increasing concern as it brings detrimental effects towards the fragile environments of Antarctica and its living organisms. Natural components including soil and water as well as a few classes of Antarctic organisms were heavily affected by diesel due to oil spill and misconduct. As the last pristine environment on Earth, extra care and precaution steps are needed to ensure that Antarctica will not become just as polluted as the rest of the world while its organisms, which are crucial to a balanced ecosystem, do not face extinction. Studies on the feasibility of bioremediation with autochthonous microorganisms are actively conducted, while several bioremediation techniques have already been trialled in the area. However, each of the methods devised comes with its own limitations. Therefore, more knowledge should be obtained especially on the on-site or off-site bioremediation approach to achieve maximum potential of microbial degradation efficiency. Furthermore, promising prospects like fungal and microalgal diesel remediation, whether independently or synergistically with bacteria, should also be studied to support the effort in contributing a new sustainable biotechnology technique.

## Figures and Tables

**Figure 1 ijerph-18-01512-f001:**
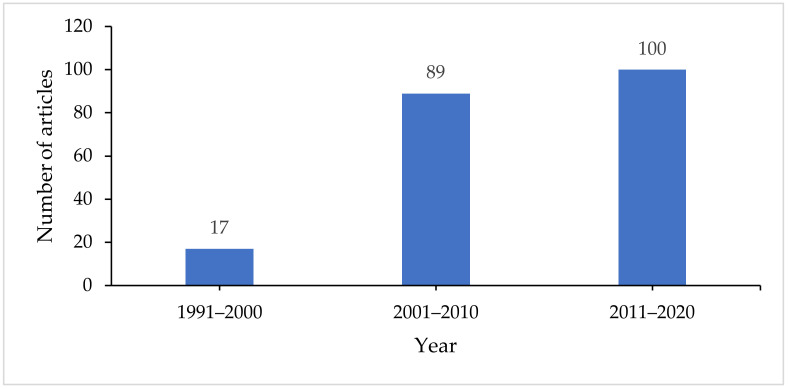
Documents published from 1991 to 2020 grouped into decades.

**Figure 2 ijerph-18-01512-f002:**
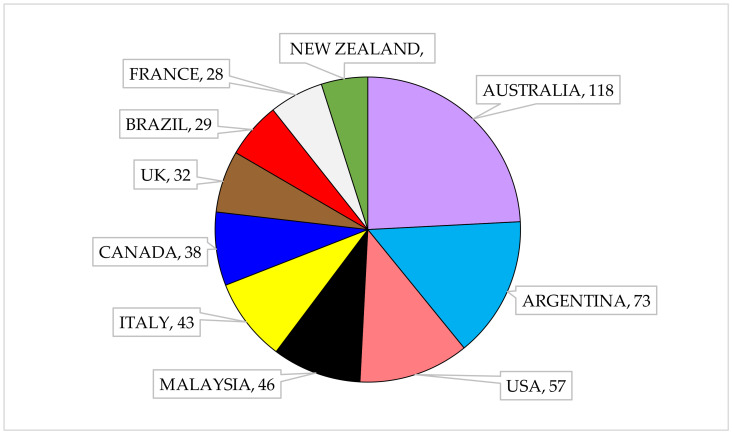
Top ten active countries in publishing Antarctic microbial hydrocarbon remediation related literature.

**Figure 3 ijerph-18-01512-f003:**
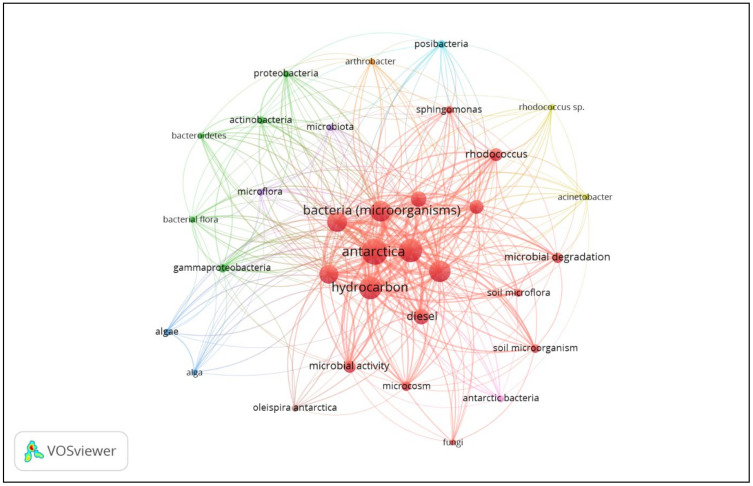
Network visualisation map of keywords in literature related to Antarctic bioremediation of diesel created by VOSviewer ©2021 Centre for Science and Technology Studies, Leiden University, The Netherlands

**Figure 4 ijerph-18-01512-f004:**
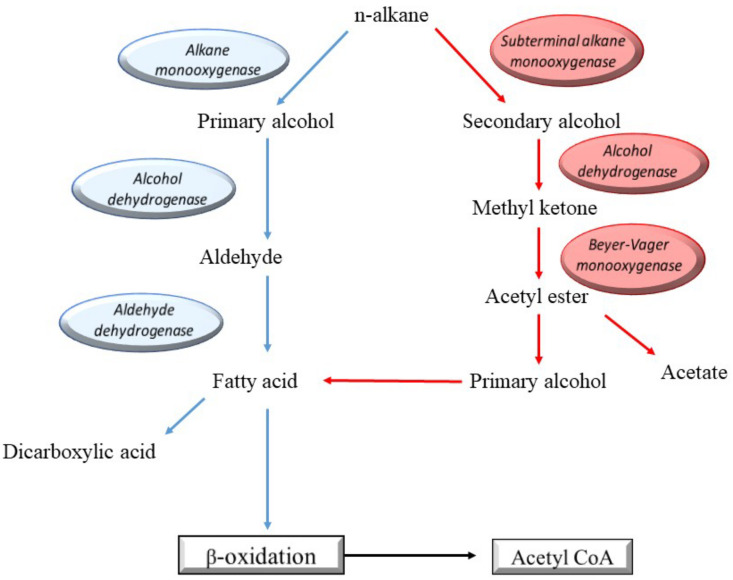
General microbial peripheral pathways of alkane degradation (adapted from [95,96]).

**Figure 5 ijerph-18-01512-f005:**
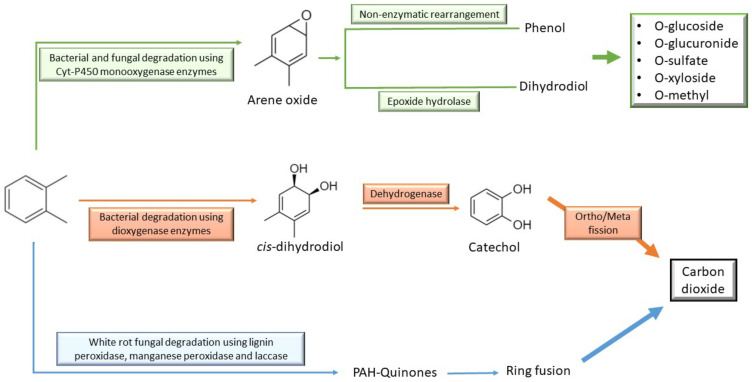
Summarised biodegradation pathway of aromatic hydrocarbons by bacteria and fungi (adapted from [92]).

**Table 1 ijerph-18-01512-t001:** Annual fuel consumption in Antarctic research stations.

Research Station	Country	Annual Fuel Consumption (L) ^1^	Alternative Energy Source (s)	Reference (s)
McMurdo Station	United States of America	5,000,000	A three-turbine wind farm was built on Ross Island, accounting for 15% of the station’s annual electricity demand. The United States Antarctic Program (USAP) has been exploring a few renewable energy sources including solar, wind, geothermal, waste wood, paper products, and recently, tidal energy.	[24,25]
Casey Station; Mawson Station; Davis Station; Macquarie Island Station	Australia	1,933,000	Two commercial-size wind turbines at Mawson station resulted on 30% fuel savings annually. A 105-panel solar array provided 10% of Casey Station total annual electricity demand.	[26,27]
Amundsen-Scott South Pole Station	United States of America	1,916,448	Wind and photovoltaic solar energy systems have been experimented and found to be very limited due to the location remoteness.	[28,29]
Syowa Station	Japan	607,000	Solar panels, solar collectors and solar hot water system provide energy for electricity and heating.	[27,30]
King Sejong Station	Republic of Korea	380,000	No reported use of renewable energy but installation of solar panels is currently being designed.	[31]
Comandante Ferraz Antarctic Station	Brazil	358,985	Integration of renewable energies and cogeneration systems have the potential to reduce annual fuel consumption.	[32]
Scott Base	New Zealand	320,000	Share the three-turbine wind farm with McMurdo Station in Ross Island, wind energy accounted to 87% of the station’s electricity demand.	[21,25,33]
SANAE IV Station	South Africa	297,872	Wind energy has the potential to meet 100% of the station’s energy demand.	[21]
Zhongshan Station	China	272,689	A power system generated by wind turbines and photovoltaic arrays was proposed but not yet commenced.	[22]
Great Wall Station	China	169,492	Solar panels were installed but were no longer operational due to damage over strong winds.	[31]
Bellingshausen Station	Russia	150,000	The station does not use renewable energies.	[31]

^1^ Annual fuel consumption data exclude the energy produced by renewable sources.

**Table 2 ijerph-18-01512-t002:** Proven or potential feasibility and limitations or bioremediation application in Antarctica.

Bioremediation Method	Feasibility and Limitations	Reference (s)
Biostimulation (nutrient amendment)	As the success of bioremediation heavily depends on its biotic and abiotic parameters, biostimulation enhances degradation by balancing the C:N:P. The addition of ad hoc nutrients will amplify the removal of hydrocarbon contaminants, therefore minimising the toxicity effect of diesel on the environment. Notwithstanding, this in-situ method might lead to the dominance of bacterial classes with the highest diesel tolerance in a community, which might affect the soil diversity in due course.	[66,67,68]
Bioaugmentation	Bioaugmentation includes the addition of pre-adapted strain to polluted site, which has been found to significantly shorten remediation period. This in-situ technique aims to maintain a high microbial biomass. Optimisation is usually performed before bioaugmentation, where actual environmental conditions, soil features, predation and competitive effects are not inferred. Moreover, the Antarctic Treaty legislates forbid any introduction or release of nonindigenous species to the area; therefore, the use of commercially available bioremediation non-Antarctic bacteria to help in cleaning up pollutants is not an option.	[65,69]
Aeration	This in-situ method was pioneered in Macquarie Island by enhancing aeration through microbioventing. This trial was conducted on the postulation that the buried oxygen sensor array would be uniform throughout the subsurface. However, buried organic and beach cobble horizons were found to limit the distribution of oxygen and nutrients in the system. In addition, a diesel-generated air compressor was required to run constantly to supply air, which hold its own environmental risks.	[1,70]
Biopile	A large-scale biopile system was successfully carried out at Casey Station and reckoned to be replicable across the Antarctic region. Biopile operation encompasses the combination of aeration via mechanical turning, fertiliser application and leachate to increase soil moisture. While this in-situ technique is promising, large-scale biopile operation is costly and requires a large amount of energy, which reduces its feasibility in remote regions. Disturbance on soil profile should also be considered.	[2,68,71]
Biocells or bioreactors	This ex-situ technique involves the containerisation of contaminated soil or water in an enclosed system with monitored biotic and abiotic parameters, thus expediting the biodegradation rate. The application of biocells or bioreactors in Antarctica is yet to be reported despite its proven practicality in several studies. However, extensive site excavation is required to relocate the contaminants to the degradation system, risking more damage to the sensitive environment.	[1,40,72,73]
Permeable reactive barriers (PRBs)	PRBs are an in-situ containment technique that is mainly used to remediate water that passes through them. It is viable in remote and cold regions due to the passive low-power system required to operate by allowing the adsorption of contaminants from flowing water to the barrier. Additionally, PRBs have the least impact on the environment since a complete removal of the system can be done at the end of site operation. Nonetheless, the viability of PRBs is limited as they do not remediate contaminated source soils.	[1,68,74]

**Table 3 ijerph-18-01512-t003:** Hydrocarbon-degrading bacterial species isolated from Antarctica.

Microbial Species	Isolation Place	Substrate Degraded	Research Summary	Reference (s)
*Arthrobacter* spp. AQ5-05	King George Island	Diesel	Up to 3% (*v*/*v*) diesel was degraded at 10 °C within 7 days Upon optimisation, metabolic activity was maximum at 16.3 °C, pH 7.67 and 1.12% (*w*/*v*) salinity.	[76,78]
*Planococcus* sp. NJ41	Antarctic Ocean	Diesel	Optimum growth at 10 °C and did not survive temperature higher than 25 °C Extracellular enzyme oxidase degraded long straight-chain hydrocarbons into shorter chains Abundant β-carotene was found in bacterial cells	[79]
*Shewanella* sp. NJ49	Antarctic Ocean	Diesel	Optimum growth at 10 °C and did not survive temperature higher than 25 °C Abundant β-carotene was found in bacterial cells	[79]
*Rhodococcus* sp. AQ5-07	King George Island	Diesel	90.39% degradation efficiency was obtained at 1% (*v*/*v*) diesel Degradation activity was optimum at 23.5 °C and 0.75% (*w*/*v*) salinity	[50]
		Waste canola oil	87.61% degradation efficiency was obtained at 3.5% waste canola oil within 3 days of incubation Optimum cellular activity occurred when incubated with 1.05 g/L ammonium sulphate and 0.28 g/L yeast extract Highest degradation rate at 12.5 °C	[64]
*Sphingomonas* sp. Ant 17	Scott Base, Ross Island	Aromatic fraction of crude oil, jet fuel and diesel	Growth was optimum at 22 °C in pH 6.4, with extended lag phase in higher pH media A great tolerance to UV irradiation and freeze-thaw cycles were displayed Genes encoding aromatic degradation enzymes are located at chromosomal level	[80]
*Pseudomonas* sp. ST41	Signy Island	Hydrocarbons	High growth observed at 4 °C in a wide range of hydrocarbons Cellular activity was the highest when bioaugmentation and biostimulation techniques were applied simultaneously	[81]
*Sphingobium xenophagum* D43FB	King George Island	Phenanthrene	Capable of assimilating diesel fuel as sole carbon source. Up to 95% phenanthrene degradation was observed. Degradation was not disrupted with the presence of cadmium as co-pollutant	[82]

**Table 4 ijerph-18-01512-t004:** Hydrocarbon-degrading fungal species isolated from Antarctica.

Fungal Species	Isolation Place	Substrate Degraded	Research Summary	Reference
*P. caribbica*	Carlini, Potter Cove, King George Island	n-alkanes and diesel fuel	Showed ability to degrade hydrocarbons and tolerance towards copper, cadmium and hexavalent chromium Production of lipase and esterase observed, proving the ability to degrade aliphatic hydrocarbons Reported as a biosurfactant producer, which relates to its ability in assimilating n-alkanes and fuel oil as carbon source	[86,87]
*Pseudeurotium bakeri*	Macquarie Island	SAB Diesel	Cell growth thrived in high concentration of SAB diesel fuel Showed hydrocarbons tolerance of up to 1000 mg/kg soil	[88]
*Mortierella* sp.	Rothera Point, Adelaide Island	Dodecane	Biomass increased when cultured on dodecane Decreased hyphal extension rate was observed, indicating dodecane as readily utilisable carbon source	[89]
*Penicillium* sp. CHY-2	Unspecified Antarctic soil	Hydrocarbons	Considerable degradation was observed in decane, butylbenzene and dodecane Manganese peroxidase (MnP), a key PAH degradation enzyme was purified	[90]
*Exophiala macquariensis*	Macquarie Island	Toluene	A novel Antarctic fungi species with toluene-degrading ability. Growth was maximum at 25 °C	[91]
*Phenoliferia glacialis*	Carlini, Potter Cove, King George Island	*n*-hexadecane	Assimilation of n-hexadecane was observed as well as cold-adapted enzyme: lipase and esterase Possessed tolerance against copper, cadmium and hexavalent chromium	[86]

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
