# Peer review of "Diesel in Antarctica and a Bibliometric Study on Its Indigenous Microorganisms as Remediation Agent"

_ijerph, 2021, doi:10.3390/ijerph18041512_

Round 1
Reviewer 1 Report
The manuscript entitled “Diesel in Antarctica and a bibliometric study on its indigenous microorganisms as remediation agent” is an review article describing the bibliometric analysis, which had demonstrated a gradual increase in number of researches on the microbial hydrocarbon remediation in Antarctica over the year. The article needs major revision before considering further. The authors need to revise the manuscript as per the following suggestions.
Special Comments
- What is the sources of diesel contaminants in Antartica please discussin the introduction at the starting lines.
- In addition to the Madrid Protocol established in 1991, there areother protocols and laws established for the contaminant removal and their impact into the environment, please add the information and discuss
- The introduction partof the review is very short and not written properly. please discuss all the literature properly to cover subject in depth. There is also a research gap need to be discussed into the revised manuscript.
- In the introduction part, only 6 references are not explaining the things properly. Please go through the detailed informationand latest review.
- Authors need to include more recent articles about the microbial bioremediation approaches with scientific depth. Thesearticles can be cited into the revised manuscript as doi; 1080/07388551.2020.1853032;doi; 10.1016/j.chemosphere.2020.127419;doi; 10.1016/j.envres.2020.109138;doi; 10.1016/j.chemosphere.2020.128827; doi;10.1016/j.envpol.2020.115908.
- Do you think the Review methodology needs to be discussed,it’s already understood to the audience? please check it.
- Revise the line 71-80.
- Please increase the data of the table-1.
- Some of the references are very old please remove them and more modern ones are needed.
- Revise the line 174-183. Please revise the sentences for clearer understanding.
- Line 197-206 please add the referenceto support.
- Describe the network visualization map more clearly in the revised version.
- Table-2 is very short please add more data about the hydrocarbon degrading bacterial strains.
- Table-3 is seems to be very short please revise with the more data from reputed journals.
- In addition to peripheral pathway, pleasediscuss the other hydrocarbon degrading bacteri
- Conclusion needs to be revised.
- Overall the manuscript needs major revision with more information. please revise the manuscript as per all the above comments.
Author Response
Comment 1
What is the sources of diesel contaminants in Antarctica please discuss in the introduction at the starting lines.
Answer: The sources of diesel contaminants in Antarctica was added in the first paragraph of introduction. (page 1; lines 37-42).
Comment 2
In addition to the Madrid Protocol established in 1991, there are other protocols and laws established for the contaminant removal and their impact into the environment, please add the information and discuss
Answer: To date, The Antarctic Treaty is the most reliable protocol in conserving the Antarctic environment. Other protocols and laws were not found, but all the parties involved in the implementation and governance of Antarctic Treaty are added. (page 2; lines 55-69)
Comment 3
The introduction part of the review is very short and not written properly. please discuss all the literature properly to cover subject in depth. There is also a research gap need to be discussed into the revised manuscript.
Answer: More information was added in the introduction part of manuscript. This includes the sources and persistence of diesel contaminants, and more discussion on the protocol on Antarctic environment conservation. (page 1-2)
Comment 4
In the introduction part, only 6 references are not explaining the things properly. Please go through the detailed information and latest review.
Answer: As more information was added, references were also added to support the information.
Comment 5
Authors need to include more recent articles about the microbial bioremediation approaches with scientific depth. These articles can be cited into the revised manuscript as doi; 1080/07388551.2020.1853032;doi; 10.1016/j.chemosphere.2020.127419;doi; 10.1016/j.envres.2020.109138;doi; 10.1016/j.chemosphere.2020.128827; doi;10.1016/j.envpol.2020.115908.
Answer: The articles doi:1080/07388551.2020.1853032 brought no result on web search. Article doi;10.1016/j.envpol.2020.115908 slightly deviates from the theme of the manuscript and was not added. However, discussion on microbial bioremediation approaches were added in Table 2. (page 6-7). The other references were added in the revised manuscript. ([62] [63] in page 6 and [101] in page 13)
Comment 6
Do you think the Review methodology needs to be discussed, it’s already understood to the audience? please check it.
Answer: Review methodology was included mainly to report the exact keywords and tools used for bibliometric analysis.
Comment 7
Revise the line 71-80.
Answer: Line 72-75 was removed in the revision. The remaining sentence structure was revised for clearer understanding of the paragraph. (page 3; lines 102-107)
Comment 8
Please increase the data of the table-1.
Answer: The amount of Antarctic research stations reporting their annual fuel consumption are limited. However, a new column was added discussing the alternative energy sources used in each station. (Table 1; page 3-4).
Comment 9
Some of the references are very old please remove them and more modern ones are needed.
Answer: The reference list was revised. Some old references were remained, but only to support unvarying information such as [58] and [59] (page 5).
Comment 10
Revise the line 174-183. Please revise the sentences for clearer understanding.
Answer: Line 174-183 was removed. It was replaced with the needs of using microorganisms in Antarctic diesel remediation and discussions on the bioremediation methods used in Antarctica (line 208-218; Table 2).
Comment 11
Line 197-206 please add the reference to support.
Answer: The information within these lines was mainly a description of Figure 2 as a part of the bibliometric analysis. Line 202-206 was meant to state the correlation between the countries involved in Antarctic Treaty and the publication analysed; not a referred information from previous research. The sentence structures within these lines were revised for clearer understanding (lines 238-241 in revised manuscript).
Comment 12
Describe the network visualization map more clearly in the revised version.
Answer: More description to the figure was added in line 249-254 in the revised manuscript. (page 8)
Comment 13
Table-2 is very short please add more data about the hydrocarbon degrading bacterial strains.
Answer: More data was added in Table 2 (Table 3 in revised manuscript) to describe more examples of hydrocarbon-degrading bacteria in Antarctica (page 9-10)
Comment 14
Table-3 is seems to be very short please revise with the more data from reputed journals.
Answer: More data was added in Table 3 (Table 4 in revised manuscript) to describe more examples of hydrocarbon-degrading fungi in Antarctica (page 10-11)
Comment 15
In addition to peripheral pathway, please discuss the other hydrocarbon degrading bacteria
Answer: Hydrocarbon-degrading microbes ware described in Table 3 and 4 (page 9-11). As this paper focuses on petrogenic hydrocarbon contaminants, other hydrocarbons were not discussed in the manuscript.
Comment 16
Conclusion needs to be revised.
Answer: Conclusion was revised with better sentence structures and fluency. Lines 366-368 were added. (page 14).
Comment 17
Overall the manuscript needs major revision with more information. please revise the manuscript as per all the above comments.
Answer: The manuscript was revised according to the comments given.

Reviewer 2 Report
The topic of this review is of great concern, as the increasing pollution level in polar regions is a very worrying issue that needs to be addressed seriously.
The paper need of a deep English revision, as many errors are present and in some sections the paper lacks of scientific sound. Moreover, I think that the manuscript needs a strong improvement. Some aspects need to be dealt with much more thoroughly. Some sections need to be completed. Furthermore, many aspects need to be clarified because there are several reasons of confusion in the manuscript. For example, despite the review should be focused on Antarctica, in many sections of the manuscript is not clear if authors are referred to Antarctic environment/organisms, or in general to marine environments. Moreover, in some cases is not clear if the authors are referred to bioremediation application in situ, or to papers focused in general on the topic (that is what I think in reality). This is a very important point. Below some suggestions:
Lines 67-69. Please, explain better. Diesel is a mixture of hydrocarbons.
Line 145. 'Other than': this expression is repeated many times in the text
I suggest to explain more deeply how the environmental parameters in polas regions influence the persistence and effects of contaminants.
4. Diesel pollution and toxicity. This section is really interesting, but I suggest reorganizing it in a more orderly way.
Line 133. "Generally, oil spill in Antarctica affects the soil biodiversity". This assumption need a reference. Are you sure that soil biodiversity is mainly affected? And if yes, which are the reasons? I suggest here to report in a ordered way the effect of oil spills on all Antarctic matrices, not only soil and soil biota. Try to give a more complete overview of the issue, addressing also acquatic environments.
Lines 163-165. This concept is not clear to me. Are you referred in general to all Antarctic invertebrates in comparison with invertebrates of temperate environments? Or only to soil invertebrate communities? And, in consideration of the fact that the authors then focus on microorganisms, I think should be important provide an overview of hydrocarbon pollution effects on microbial communities in Antarctica. See also:
Polar Biol (2015) 38:1565–1574, DOI: 10.1007/s00300-015-1717al-9;
Microorganisms 2019, 7, 632; doi:10.3390/microorganisms7120632.
Line 179. Provide references of bioremediation techniques applied in Antarctica, if any.
Line 180. This is in contrast with lines 176-177.
Lines 184-186. Is this true for Antarctica?
Lines 188-190. "...seen indicating the growing attention of microbial hydrocarbon remediation in the South Pole region...". I suggest to explain exaclty what you mean. You mean paper focused on....? Isolation of hydrocarbon-degrading bacteria? On the hydrocarbon remediation potential of cold-adapted bacteria? On the effective application of bioremediation techniques? In vitro or in situ?
Line 222. at low temperature
Lines 222-223. Please, explain better. This sentence in principle means that bacteria are able to degrade hydrocarbons thanks to beta-carotene, while in reality is and indirect effect.
Line 226. The name of genes and bacterial genera should be in italics
Line 229. Please delete 'as'
Line 236. I think that the word hydrocarbon should be in plural. Please check throughout the manuscript.
Line 248. Please correct, the correct verb is were
Lines 253-260. These are in general adaptative strategies to cold environment conditions. I suggest to re-structure all this part. I think is better to provide a separate section with all adaptations of microorganisms to live in cold environments, for bacteria, fungi and microalgae. Then, the authors should contextualize and explain how all of these adaptations could be related to the hydrocarbon degrading potential.
Line 272. Are you referred to Antarctic bacterial species? Please specify.
Line 279. Aromatic hydrocarbonS ARE.
Paragraph 6. Metabolic pathways of bacterial diesel degradation. This section is referred only to bacteria, but the review deals also about other microorganisms. Thus, I suggest to improve this section with proper references also for the fungi and microalgae.
In my opinion, the manuscript deserves publication, but only after strong efforts for improvements.
Author Response
Comment 1
Lines 67-69. Please, explain better. Diesel is a mixture of hydrocarbons.
Answer: Lines 67-69 were removed and replaced with lines 95-100 in the revised manuscript. (page 2-3).
Comment 2
Line 145. 'Other than': this expression is repeated many times in the text.
Answer: Other coordinating conjuctions were used interchangeably in the revised manuscript.
Comment 3
I suggest to explain more deeply how the environmental parameters in polas regions influence the persistence and effects of contaminants.
Answer: More discussion on the persistence and toxicity effect of diesel contaminants was added in the introduction and subtopic 4 (lines 157-180; page 5).
Comment 4
Diesel pollution and toxicity. This section is really interesting, but I suggest reorganizing it in a more orderly way.
Answer: This section was revised to improve fluency and readers’ understanding.
Comment 5
Line 133. "Generally, oil spill in Antarctica affects the soil biodiversity". This assumption need a reference. Are you sure that soil biodiversity is mainly affected? And if yes, which are the reasons? I suggest here to report in a ordered way the effect of oil spills on all Antarctic matrices, not only soil and soil biota. Try to give a more complete overview of the issue, addressing also acquatic environments.
Answer: Line 133 "Generally, oil spill in Antarctica affects the soil biodiversity" was removed. The discussion of diesel toxicity effect was broadened in this revision. Diesel toxicity on soil and marine environments was discussed side to side. However, literatures on the toxicity effect on marine environment were little as compared to soil environment (page 5).
Comment 6
Lines 163-165. This concept is not clear to me. Are you referred in general to all Antarctic invertebrates in comparison with invertebrates of temperate environments? Or only to soil invertebrate communities? And, in consideration of the fact that the authors then focus on microorganisms, I think should be important provide an overview of hydrocarbon pollution effects on microbial communities in Antarctica. See also: Polar Biol (2015) 38:1565–1574, doi:10.1007/s00300-015-1717al-9; Microorganisms 2019, 7, 632, doi:10.3390/microorganisms7120632.
Answer: These lines were removed. In the revised manuscript, description of soil invertebrates and marine invertebrates were stated clearly to avoid confusion. Toxicity effect on microbial community upon hydrocarbon pollution was added. The suggested references were analysed and added into the revised manuscript ([13] in page 2 and [57] in page 5).
Comment 7
Line 179. Provide references of bioremediation techniques applied in Antarctica, if any.
Answer: Bioremediation techniques applied in Antarctica was discussed in Table 2 (page 6-7).
Comment 8
Line 180. This is in contrast with lines 176-177.
Answer: Lines 175-187 were removed as the sentence structures were significantly flawed. In the revised manuscript, this paragraph was replaced with lines 208-218 (page 6).
Comment 9
Lines 184-186. Is this true for Antarctica?
Answer: These lines were removed. Bioremediation techniques applied in Antarctica was discussed individually in Table 2 (page 6-7).
Comment 10
Lines 188-190. "...seen indicating the growing attention of microbial hydrocarbon remediation in the South Pole region...". I suggest to explain exaclty what you mean. You mean paper focused on....? Isolation of hydrocarbon-degrading bacteria? On the hydrocarbon remediation potential of cold-adapted bacteria? On the effective application of bioremediation techniques? In vitro or in situ?.
Answer: The statement is further justified to explain the data of the graph. The bibliometric analysis result was generated based on the keywords query mentioned in Section 2. It is not narrowed down into specific topic, but rather a broad scope within the keyword coverage. Hence, it cannot pinpoint the growing attention in a certain focus.
Comment 11
Line 222. at low temperature.
Answer: This language error was amended.
Comment 12
Lines 222-223. Please, explain better. This sentence in principle means that bacteria are able to degrade hydrocarbons thanks to beta-carotene, while in reality is and indirect effect.
Answer: This sentence was flawed and misleading. Beta-carotene explains the adaptation of bacteria in low temperature. This information was moved to a new subtopic (7. Microbial adaptation mechanisms in cold region) as it has no direct effect to diesel degradation. (page 13-14).
Comment 13
Line 226. The name of genes and bacterial genera should be in italics.
Answer: This formatting error was amended.
Comment 14
Line 229. Please delete 'as'.
Answer: This grammatical error was amended. The language of the entire manuscript was revised.
Comment 15
Line 236. I think that the word hydrocarbon should be in plural. Please check throughout the manuscript.
Answer: The entire manuscript was revised to amend this language error. Some “hydrocarbon” was remained in singular form when used as a phrase. For example, “hydrocarbon remediation”; “hydrocarbon contaminants”. .
Comment 16
Line 248. Please correct, the correct verb is were
Answer: This grammatical error was amended.
Comment 17
Lines 253-260. These are in general adaptative strategies to cold environment conditions. I suggest to re-structure all this part. I think is better to provide a separate section with all adaptations of microorganisms to live in cold environments, for bacteria, fungi and microalgae. Then, the authors should contextualize and explain how all of these adaptations could be related to the hydrocarbon degrading potential.
Answer: The microbial adaptive strategies were rewritten in a newly added subtopic 7 (page 13-14). Several more information was added, but literatures on the correlation of the adaptations to their hydrocarbon-degrading ability was limited.
Comment 18
Line 272. Are you referred to Antarctic bacterial species? Please specify.
Answer: The sentence was referring to bacteria species in general, where references used was specifying to temperate region. The sentence was amended for specification.
Comment 19
Line 279. Aromatic hydrocarbonS ARE.
Answer: This grammatical error was amended.
Comment 20
Paragraph 6. Metabolic pathways of bacterial diesel degradation. This section is referred only to bacteria, but the review deals also about other microorganisms. Thus, I suggest to improve this section with proper references also for the fungi and microalgae.
Answer: This section was amended to include discussion on fungal and microalgal degradation mechanism. Figure 5 was removed, and figure 6 was replaced to describe bacterial and fungal degradation together (Figure 5 in revised manuscript; page 13) whereas the mechanism of microalgal was not yet documented, as stated in the revised manuscript.

Round 2
Reviewer 1 Report
The manuscript has been improved and can be accepted for publication according to the revisions carried out.